# Prediction of *Klebsiella* phage-host specificity at the strain level

**Dimitri Boeckaerts** [1,2], **Michiel Stock** [2], **Celia Ferriol-González** [3], **Jesús Oteo-Iglesias**[4,5], **Rafael Sanjuán** [3], **Pilar Domingo-Calap** [3], **Bernard De Baets** [2] **& Yves Briers** [1] ✉

Phages are increasingly considered promising alternatives to target drug-resistant bacterial pathogens. However, their often-narrow host range can make it challenging to find matching phages against bacteria of interest. Current computational tools do not accurately predict interactions at the strain level in a way that is relevant and properly evaluated for practical use. We present PhageHostLearn, a machine learning system that predicts strain-level interactions between receptor-binding proteins and bacterial receptors for *Klebsiella* phage-bacteria pairs. We evaluate this system both in silico and in the laboratory, in the clinically relevant setting of finding matching phages against bacterial strains. PhageHostLearn reaches a cross-validated ROC AUC of up to 81.8% in silico and maintains this performance in laboratory validation. Our approach provides a framework for developing and evaluating phage-host prediction methods that are useful in practice, which we believe to be a meaningful contribution to the machine-learning-guided development of phage therapeutics and diagnostics.

Phages, bacterial viruses, are among Earth's most abundant viruses[1]. They typically have a limited host range at the strain level[2,3], although broad host range phages infecting multiple species have also been described[4,5]. For this reason, phages and the proteins they encode have the potential to become precise therapeutics and diagnostics that can target (multidrug-resistant) bacteria[6]. However, finding matching phages against specific hosts of interest can be challenging in ecological and therapeutic settings[7,8].

In recent years, novel computational tools that predict interactions between phages and their potential hosts have tried to overcome this bottleneck[8,9]. Most of these tools are based on measuring the similarity between a query phage genome and potential host genomes or exploiting the similarity between the query phage genome and other phage genomes (*e.g.*, codon usage) with known hosts[10]. Two such approaches are iPHoP and CHERRY. iPHoP is a two-stage machine

learning framework developed by Roux et al.[10]. It integrates multiple existing methods to make host predictions at the genus level for a broad range of phages, with the goal of maximizing correct host predictions for metagenome-assembled viruses. This framework attains a low false discovery rate (<10%). CHERRY uses a graph-based deep learning model that predicts hosts at the species level by incorporating multiple types of interaction information (*e.g.*, genome sequence similarity, CRISPR signals, and others) in a multimodal graph[11]. Interestingly, CHERRY can predict new interactions starting from a virus query as well as from a prokaryote query. As Roux et al.[10] argue, predictions at the genus or species level are essential within the context of viral ecology. In clinical applications, however, knowing a phage's specificity at the strain level is typically desired[12–14], which remains a bottleneck for developing phage therapeutics and diagnostics. Practically, this could be overcome by developing models that predict

[1]Laboratory of Applied Biotechnology, Department of Biotechnology, Ghent University, Ghent, Belgium. [2]KERMIT, Department of Data Analysis and Mathematical Modelling, Ghent University, Ghent, Belgium. [3]Institute for Integrative Systems Biology (I2SysBio), Universitat de Valencia-CSIC, Paterna, Spain. [4]Laboratorio de Referencia e Investigación en Resistencia a Antibióticos e Infecciones Relacionadas con la Asistencia Sanitaria, Centro Nacional de Microbiología, Instituto de Salud Carlos III, Madrid, Spain. [5]CIBER de Enfermedades Infecciosas (CIBERINFEC), Instituto de Salud Carlos III, Madrid, Spain. ✉e-mail: Yves.Briers@UGent.be

phage-host interactions and make these actionable. For example, prediction scores can be used to prioritize which phage-host combinations should be tested in the laboratory, reducing labor-intensive work to a minimal set of predicted top candidates to be validated. Correspondingly, the overall tool should also be evaluated in a manner that is representative of practical use.

Protein language models are increasingly popular frameworks for machine learning applications at the protein level[15–17]. These are state-of-the-art deep learning models that are trained on large amounts of protein sequence data in a self-supervised manner. Specifically, the models learn to predict the occurrence of amino acids in the context of other amino acids. By training at an enormous scale, these models effectively learn the underlying distribution of naturally occurring proteins. A trained protein language model can be used to generate accurate numerical representations for proteins or be further tweaked for specific problems using a much smaller dataset, a process called fine-tuning[15]. As a result, these large models remove the need for explicit feature engineering and allow for an end-to-end approach.

We have previously proposed a general, biology-informed multi-layer machine learning approach to elucidate phage-host interactions at the strain level by making predictions at each stage of the replication cycle[18]. In that approach, the first layer represents the initial interaction between the phage receptor-binding proteins (RBPs) and the bacterial surface receptors. Typically, RBPs constitute the primary determinant of host specificity[19]. The study by Sørensen et al.[2] aligns well with this first layer. The authors computationally analyzed tail spike protein diversity in 99 *Ackermannviridae* phages to determine phage host specificity at the level of the interacting O-antigen receptors, effectively at the strain level. However, this approach is not geared towards applications in a clinical context, nor is it built into a tool that other researchers can easily use.

In the present study, we develop and validate a new machine learning approach called PhageHostLearn, which predicts the initial interactions between RBPs and bacterial receptors for *Klebsiella* phage-bacteria pairs at the strain level (Fig. 1a). PhageHostLearn is developed as the first layer in the proposed multilayer machine learning approach. *Klebsiella pneumoniae* is among the most prominent multidrug-resistant pathogens worldwide[20]. The unique public availability of interaction data for *Klebsiella* phage-host interactions enables the development of prediction methods at the strain level, a remaining bottleneck for most other bacterial species[21]. *Klebsiella* bacteria produce capsule polysaccharides (CPS), also termed K-antigens, which comprise sugar molecules that are present in repeats and form a protective layer on the bacterial surface[22]. For most *Klebsiella* phage-host interactions, the initial interaction consists of phage RBPs recognizing the CPS and is typically the most important determinant of host specificity[3,23]. Therefore, PhageHostLearn processes phage and bacterial genomes into phage RBPs and bacterial K-locus proteins, respectively. We use the ESM-2 protein language model to encode protein sequences as numerical vector representations, which are used as input for an Extreme Gradient Boosting (XGBoost) model, a widely used and broadly applicable method[24]. PhageHostLearn allows predicting interactions in both directions (*i.e.*, from phage to bacterium and vice versa), which is a typical example of pairwise learning[25]. Furthermore, our approach outputs a ranking of potential phage candidates for in vitro validation of a given bacterium. We thoroughly evaluate this approach both in silico and in vitro, in the clinically relevant setting of finding matching phages against a new bacterial strain (Fig. 1b, c). We show that PhageHostLearn reaches a cross-validated ROC AUC up to 81.8% in silico and can hold on to this performance in the laboratory. Our approach is made publicly available as a tool that can be further improved over time. We believe this to be a meaningful first step in the machine-learning-guided development of phage therapeutics and diagnostics.

## Results

### Sequence data collection and processing

Phage genome sequence data, bacterial genome sequence data and their in vitro verified phage-bacteria interactions were collected from the Institute for Integrative Systems Biology (I²SysBio) in Valencia, Spain as described by Beamud et al.[3] and Ferriol-González et al.[26]. In total, a diverse set of 105 phage genome sequences and 200 bacterial genome sequences were collected. Spot tests were performed to test for phage-bacteria interactions. Out of 10,006 spot tests performed in total, 333 are confirmed interactions (3.33%). Interactions are considered positive if a spot is visible using a 1:10 phage dilution, reflecting an initial interaction between phage RBPs and host receptors (but not necessarily a productive replication).

Phage genome sequences were processed in three consecutive steps (Fig. 1a): (1) PHANOTATE was used to identify genes in each of the phage genomes (McNair et al.,[27]); (2) phage RBPs were detected among the translated protein sequences of the identified genes, following our method outlined in Boeckaerts et al.[28]; and (3) detected phage RBPs shorter than 200 amino acids and longer than 1500 amino acids were discarded, according to the range in length in which we expect RBPs[29]. In total, 9727 genes were detected with PHANOTATE, and subsequently, 274 phage RBPs were detected among those identified genes. We detected at least one RBP in each phage genome, and up to eight RBPs in a single phage genome (Supplementary Fig. S1).

Bacterial genome sequences were processed with Kaptive[30,31] to identify the capsule synthesis locus (K-locus) in each of the bacterial genomes (Fig. 1a). On average, each K-locus consisted of 19 proteins that constitute the K-antigen (the number of proteins was between 10 and 25). A total of 89 different KL-types were identified using Kaptive (out of a current estimate of 134 KL-types[30]), representing a significant diversity in *Klebsiella* strains. The KL13 type was most often represented, while 45 different KL-types were only represented once (Supplementary Fig. S2).

### Multi-instance feature representations

We transformed phage RBPs and bacterial K-locus proteins into combined numerical vector representations (so-called joint features), to serve as input to the machine learning model (Fig. 1a). These representations are so-called multi-instance representations[32], combining one or multiple RBPs per phage and multiple K-locus proteins per bacterium.

We used the pre-trained ESM-2 protein language model (t33_650M_UR50D configuration) to transform each of the RBPs and K-locus proteins into a unique numerical vector[33]. The vectors corresponding to the RBPs of the same phage or the K-locus proteins of the same bacterium were averaged into multi-instance representations for each phage or bacterium. Finally, for each known interaction in the dataset, the multi-instance representations of each phage and each bacterium were concatenated into a final combined numerical vector that represents a known phage-host pair.

### A classification model that predicts interactions

We trained a binary XGBoost classifier to output prediction scores reflecting how likely a phage-host pair will interact based on the combined ESM-2 numerical vector representations described above (Fig. 1a). The maximum depth of each tree, the learning rate, and the number of estimators were tuned using stratified five-fold cross-validation (Table 1). The optimal maximum depth was 7, the optimal learning rate was 0.3 and the optimal number of estimators was 250.

### In silico evaluation of the model

We have evaluated our model both in silico and in the laboratory in the practical setting of finding which phages in the collection are the most active against a given bacterial strain. A predictive model is useful if it can effectively suggest the most appropriate phages to test, in that way

### a. PhageHostLearn system

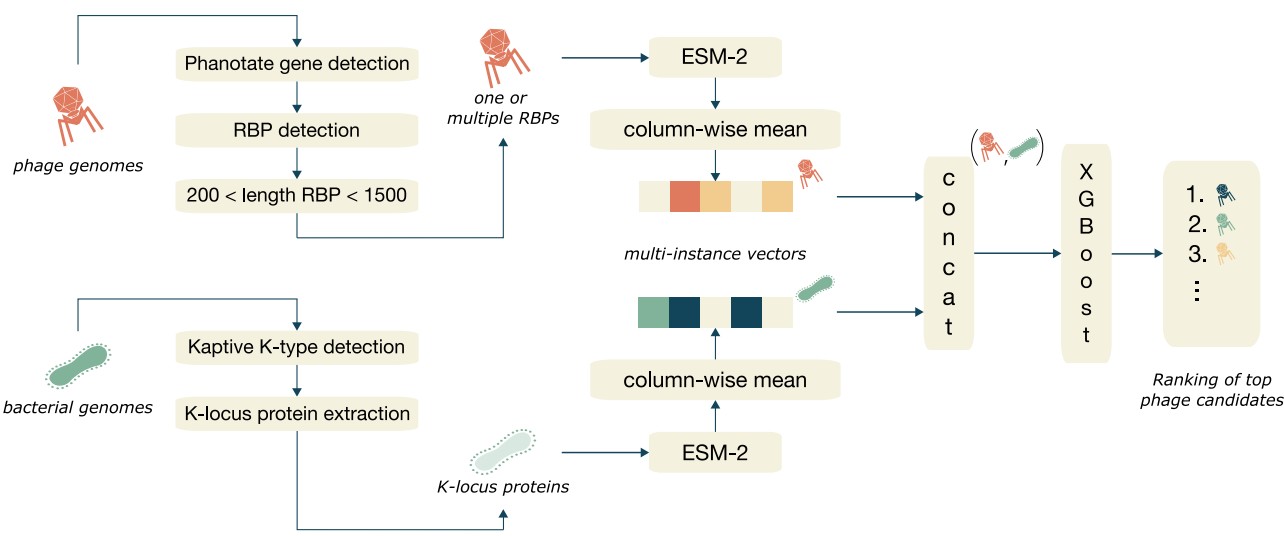

### b. *In silico* validation with LOGOCV

### c. *In vitro* validation with spot tests

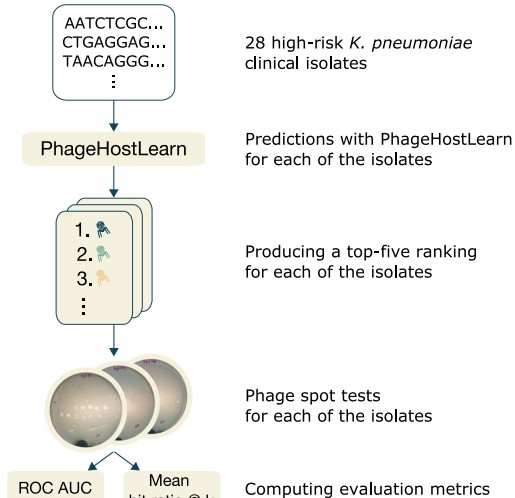

**Fig. 1 | PhageHostLearn overview and validation procedures. a.** Overview of the PhageHostLearn machine learning system. PhageHostLearn processes phage and bacterial genomes into phage RBPs and bacterial K-locus proteins, respectively. Phage RBPs belonging to the same phage and bacterial K-locus proteins belonging to the same bacterium are combined into separate multi-instance representations using ESM-2. These multi-instance representations are concatenated into combined representations of the phage-host pairs. Finally, these representations are given as input into an XGBoost model to make predictions and output a ranking of top candidate phages to test against a given bacterium. **b.** In silico validation of the PhageHostLearn system using a leave-one-group-out cross-validation (LOGOCV) scheme that measures the ROC AUC and mean hit ratio @ *k* as evaluation metrics. **c.** In vitro validation of the PhageHostLearn system using 28 high-risk *K. pneumoniae* clinical isolates in Spain. The PhageHostLearn system predicts a top-five ranking for each of the clinical isolates. For each ranking, the top five phage candidates are validated in the laboratory using phage spot tests.

minimizing manual analysis and laborious experimental work. We have simulated this representative setting in silico by iteratively holding out a single or group of the bacterial genome(s) with its phage interactions

### Table 1 | Hyperparameters and their tested values in the PhageHostLearn model

| Hyperparameters | Tested values |
|---|---|
| Maximum depth | 3, 5, **7**, 9 |
| Learning rate | 0.2, **0.3**, 0.4 |
| Number of estimators | **250**, 500, 750 |

The optimal values of the hyperparameters for the model are indicated in bold.

at a time from the training set. Pairwise identity scores between K-locus nucleotide sequences were used to construct groups at set thresholds ranging from 75% to 100%. In each iteration, the held-out interactions were predicted by the model and their prediction scores were used to construct rankings of the predicted phages. The hit ratio, defined as the probability of finding at least one matching phage, was computed across the top-*k* ranked phages by comparing the ranked predictions to the ground truth labels to quantify how well our model finds matching phages. This process was repeated for values of *k* ranging from 1 to the total number of phages, was repeated for each of the grouped bacterial genomes in the dataset, and finally averaged across all the iterations over the grouped bacterial genomes (Fig. 2a). This mean hit ratio @ *k* provides a meaningful visualization of the average

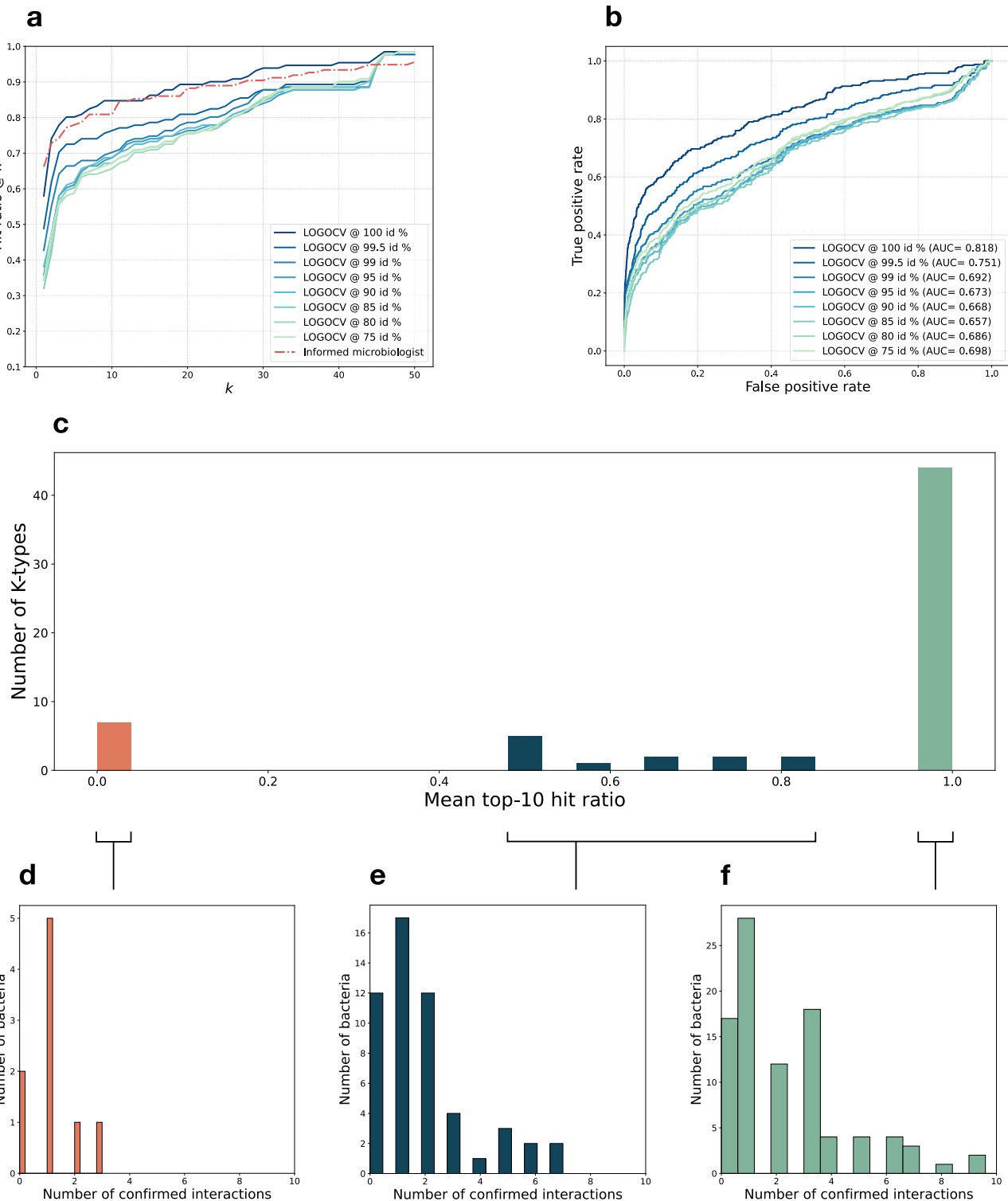

**Fig. 2 | In silico validation results of PhageHostLearn. a.** Mean hit ratio @ $k$ of the trained XGBoost model in a LOGOCV at decreasing thresholds for K-locus identity (blue-green curves) and of an informed microbiologist approach (red). At the 100% threshold for grouping, identical K-locus sequences are grouped together, either in the training set or test set. **b.** ROC curve with AUC of the trained XGBoost model in a LOGOCV at decreasing thresholds for K-locus identity. **c** Histogram of the mean top-10 hit ratio against the number of KL-types for which that hit ratio was achieved. There is a contrast between KL-types that are perfectly predicted (hit ratio is 100%) and not at all predicted (hit ratio is 0%). **d–f** Histograms of the number of confirmed interactions per bacterial strain related to the KL-types with a mean top-10 hit ratio of respectively 0%, 50–80%, and 100%.

probability of finding at least one hit in the top-$k$ candidates suggested by the model. For example, with our model, we expect to find at least one hit in the top-10 in around 65% to 84% of the cases on average. The model's performance decreases as it makes predictions for bacterial strains that are increasingly dissimilar from those seen in training (blue-green curves) and stabilizes at thresholds around 90% identity, a typical threshold for defining unique KL-types[31]. Given a majority of KL-type specific phage-host interactions in our dataset, we presume that

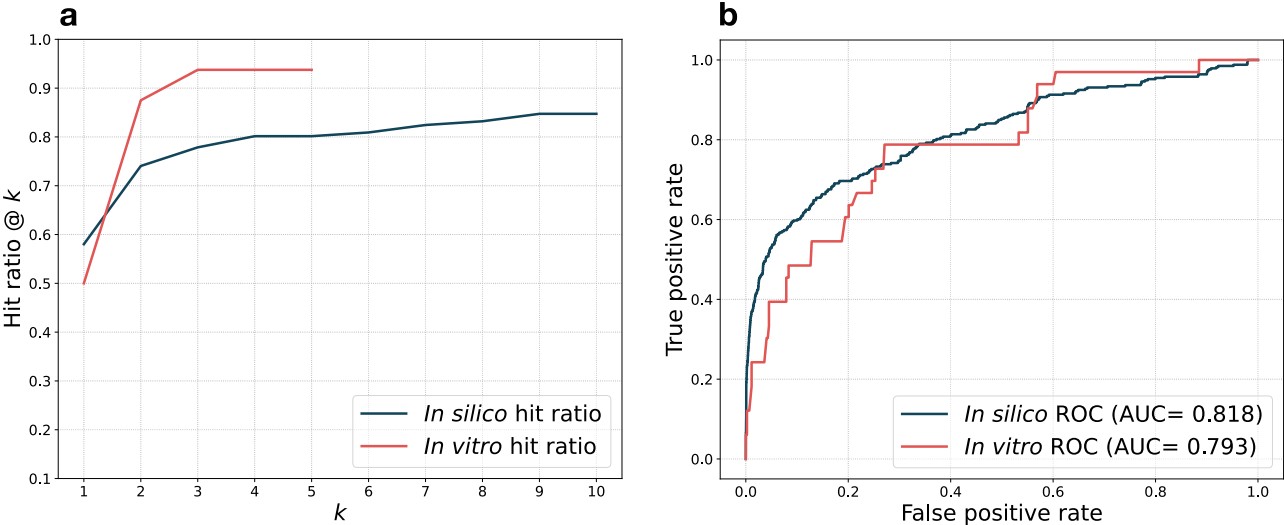

**Fig. 3 | Comparison of in silico and in vitro validation results of PhageHostLearn. a.** Mean hit ratio @ *k* comparing the in silico validation and the in vitro validation of the XGBoost model. **b.** ROC curve with AUC comparing the in silico validation and the in vitro validation of the XGBoost model.

PhageHostLearn learns some robust patterns that are KL-type independent, but currently overfits on KL-type specific patterns (which can be observed by the decrease in performance from 100 to 90% identity). We have also simulated an informed microbiologist approach by manually selecting from a subset of phages that are known to infect the same KL-type as the bacterial strain at hand (red curve). PhageHostLearn can reach a performance comparable to that of an informed microbiologist at the highest threshold for grouping, however, given the decrease in performance for only slightly lower thresholds, seems to rely on very similar K-locus sequences to reach that level of performance. In addition, we visualize the receiver operating characteristic (ROC) curves (Fig. 2b) and measure their area-under-the-curve (AUC) as a general performance metric of our model. The ROC AUC can be interpreted as the probability that the model will score a randomly chosen interacting phage-host pair higher than a randomly chosen phage-host pair that does not interact. Our model reaches ROC AUC values between 65.7% and 81.8%. Expectedly, the mean hit ratio differs across different KL-types, and there is a strong contrast between the top-10 mean hit ratio for the best and worst predicted KL-types (Fig. 2c). Therefore, we constructed histograms of the number of confirmed interactions per bacterial strain belonging to the best and worst predicted KL-types and the group in between. We observed that the performance across KL-types can be related to the number of confirmed interactions in those KL-types (Fig. 2d–f), highlighting the need for an extensive training dataset with sufficient confirmed interactions for each KL-type for optimal performance.

**In vitro validation of the model with spot tests**

A total of 28 carbapenem-resistant *K. pneumoniae* clinical isolates were collected and sequenced in collaboration with the National Micro-biology Center (CNM) in Madrid, Spain. These *K. pneumoniae* clinical isolates comprised high-risk clones that are currently circulating in Spain and included a total of eight different KL-types (KL17, KL24, KL27, KL64, KL102, KL107, KL112, and KL151). Each of these KL-types was also present (at least once) in the training data. For each of these *K. pneumoniae* clinical isolates, PhageHostLearn was used to predict interactions and construct a ranking for the collection (I²SysBio) of 59 phages isolated on *Klebsiella* spp. reference strains and for which the full genome was available[26]. As these phages were isolated on *Klebsiella* spp. reference strains, they were not tested on all the KL-types present in the test set of clinical isolates. Moreover, none of the phages were tested before on these specific clinical isolates. The top-five ranked

phages for each *K. pneumoniae* clinical isolate were validated in the laboratory using spot tests at 1:10 and 1:10³ phage dilutions in duplicate or triplicate (for discrepant results). Spot tests were used for consistency across model training and in vitro validation and because we focused on the initial interaction between phage RBPs and host receptors (not necessarily reflecting a productive replication). In an additional effort, all 17 unique phages that were identified across the different top-five lists, were tested against all the 28 clinical isolates to examine potential false negatives. Finally, DefenseFinder was used to find phage defense systems in the isolates' genomes[34].

One or more interactions were confirmed with spot tests for 16 out of the 28 bacterial isolates. Out of these 16 isolates, one or more interactions could be confirmed at both dilutions for 12 of the isolates (Supplementary Datasets 1–3). Said differently, across all the interactions we tested, 33 could be confirmed at a 1:10 phage dilution and 16 of those could also be confirmed at a 1:10³ phage dilution. These differences between the two dilutions may indicate the presence of one or more active phage defense systems (due to the occurrence of lysis without successful replication at a 1:10 phage dilution). In total, we have found 43 unique phage defense systems using DefenseFinder (Supplementary Datasets 5). PhageHostLearn could correctly predict hits in the top-five phage candidates for 15 of these isolates, corresponding to a top-five hit ratio of 93.8% (Fig. 3a). Comparing the different KL-types, PhageHostLearn only missed 7 hits in total, for strains of KL17, KL24 and KL27 (Supplementary Datasets 4). Overall, the PhageHostLearn system retains its in silico performance, reaching a ROC AUC of 79.3% in this in vitro validation, compared to up to 81.8% in silico (Fig. 3b).

Overall, the top candidates predicted by the XGBoost model are often phages that have a broader host range, such as K65PH164, K30λ2.2, K2064PH2, and K7PH164C4 (Supplementary Datasets 1). These phages appear across the true positives, false positives, and false negatives (Table 2). Considering that these phages are not K-locus specific, this result was to be expected based on our focus on RBPs and K-locus proteins. Interestingly, the model often suggests a strategy that a microbiologist would think also think of: testing all the broad host range phages by default. The model also correctly suggests KL-type specific phages (*e.g.*, K54λ1.1.1 and K17α62) for five out of the six clinical strains for which KL-type specific phages could be confirmed experimentally (Supplementary Datasets 1). One KL-type specific phage (K17α61) was wrongly predicted as a false positive in combination with some bacterial strains but was a false negative prediction in

**Table 2 | Concordance of the predictions by our model with laboratory confirmations by means of a confusion table**

| Case | Model prediction | In vitro result | Count | Prevalent phages |
|---|---|---|---|---|
| True positive | Top-five | Interaction | 26 | K65PH164, K30λ2.2, K2064PH2, K7PH164C4 |
| False positive | Top-five | No interaction | 114 | K65PH164, K2064PH2, K29PH164C1, K17α61 |
| False negative | Outside of top-five | Interaction | 7 | K2064PH2, K7PH164C4, K17α61, K30λ2.2 |
| True negative | Outside of top-five | No interaction | Not considered | – |

Counts of true positives, false positives, and false negatives with the most prevalent phages in each category across all the top-five laboratory-confirmed interactions (140 in total), supplemented by the interactions tested in all 17 unique phages across the different top-five lists for counting the false negatives. The true positives were the predictions in the top-five recommendations that were confirmed in the in vitro validation. The false positives were the predictions in the top-five recommendations that could not be confirmed in the in vitro validation. Finally, the false negatives were the interactions that could be confirmed in the lab across the 17 tested phages that were not predicted in the top-five recommendations.

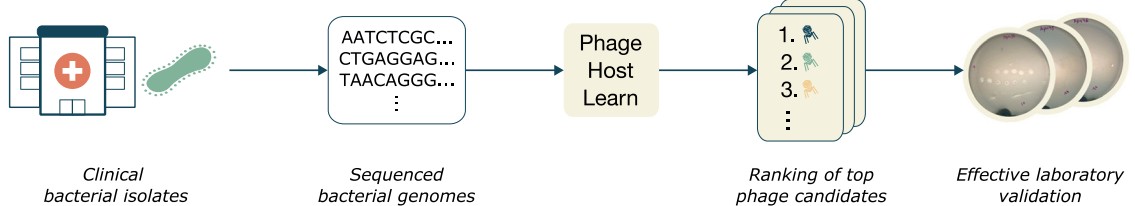

**Fig. 4 | PhageHostLearn can guide effective laboratory validation of clinical bacterial isolates that are sequenced.** The system produces prediction scores that are used to construct a ranking of phage candidates, which is an actionable format from which laboratory validation can be focused on the top-$k$ ranked phages.

combination with other bacterial strains. These wrong predictions are more challenging to explain from a biological perspective and could equally be explained because of a lack of sufficiently similar data from which the models can learn.

Moreover, the model correctly suggested 78.8% (= 26 / [26 + 7]) of all the confirmed interactions in the top-five. However, these seven false negatives can be an underestimation, as we have not comprehensively tested all 59 phages against the 28 clinical isolates. In addition, the model also suggested 114 interactions that turned out to be negative. This is intrinsic to using a ranking approach because top suggestions are tested regardless of the prediction scores that are assigned by the model, thus could also comprise phages that do not adsorb to the host strain. Concretely, when all interactions for a given bacterium received low scores, the five with the highest scores were still tested.

## Discussion

In this work, we developed PhageHostLearn, a machine learning system that overcomes three current bottlenecks for phage-host interaction prediction in the context of phage therapeutics and diagnostics. First, the system predicts phage-host interactions at the strain level. Second, it outputs prediction scores that can be used to recommend top candidates, resulting in a more effective laboratory validation. Third, we evaluated this system in the practical setting it will be used for, i.e., predicting matching phages for new bacteria.

We specifically trained and evaluated our system to make predictions for *Klebsiella* phage-host pairs in an actionable format. The unique data availability for *Klebsiella* allowed us to construct a machine learning system capable of making predictions at the strain level. Moreover, we have deliberately chosen to focus on phage RBPs and bacterial K-locus proteins, as these proteins are involved in the first step of the phage infection cycle and known to be a very determining factor of phage-host specificity for many *Klebsiella* phages[3,23,35]. We hypothesize that the same approach could be extended to predict phage-host interactions with similar biological characteristics at the level of phage RBPs and host receptors. For example, *Escherichia coli*, *Salmonella enterica* and *Acinetobacter baumannii* all have characteristic O-antigens that many phages bind to through their RBPs[36,37]. However, host-specific diversity (*e.g.*, over 180 different *E. coli*

O-antigen serotypes compared to around 130 different *Klebsiella* K-types) might be an additional challenge and might require larger amounts of data[38]. In contrast, our approach might be less relevant for phages where specificity is less determined at the strain level (*e.g.*, *Staphylococcus* phages)[39]. PhageHostLearn can also be extended to include other typical phage receptors, such as outer membrane proteins, flagella, and others, given that they can be annotated in the genome.

A combined, multi-instance feature representation was computed using the ESM-2 protein language model (Fig. 1a). This way of computing features is inspired by how state-of-the-art deep learning architectures process natural language into numerical representations. The advantage of this type of approach is that it bypasses the need for explicit feature engineering, such as computing codon usage or $k$-mer frequencies, which is seen in many earlier approaches. These methods result in combined multi-instance representations that represent the phage-host pairs together. The XGBoost model, on top of these combined multi-instance representations, outputs prediction scores that can be used to propose top phage-host candidates to test. We show that our machine learning system suggests top candidates better compared to a typical microbiologist approach. However, we hypothesize that there may be better ways of aggregating individual ESM-2 protein representations into multi-instance representations compared to our simple approach of computing a column-wise mean.

PhageHostLearn produces prediction scores that can be used to rank top phage candidates for a given bacterium, a practical output format that is directly actionable and can guide effectively in vitro validations (Fig. 4). At the same time, a ranking removes the need to set an arbitrary cutoff on the prediction score above which predictions are interpreted as interacting phage-host pairs. The ranking is closely linked to the way we evaluate our model in silico and in vitro: by instructing the model to predict interactions of given phages against a new bacterium and evaluating to what extent the model is useful in assigning higher prediction scores to matching phages (effectively resulting in a ranking that is useful in practice). While both the in silico and in vitro evaluations provide evidence of the model's accuracy and robustness, a larger and more diverse set of clinical isolates for the in vitro evaluation is expected to further increase the robustness of the evaluation. The current in vitro hit ratio @ $k$ was higher than the in

silico hit ratio @ *k*, but we would expect the in vitro curve to move down towards the in silico curve as the number of isolates to test on increases. In addition, we notice that such a ranking can include (many) false positives, an inherent trait of a ranking approach, given that we expect few interacting phage-host pairs overall for *Klebsiella* phage-host pairs. Only 3.33% of the interactions in the training data could be confirmed, which translated to finding one hit among every 30 phages. In a practical setting, we argue that avoiding false negatives is, up to a certain extent, more important than avoiding false positives. Two strategies can be further explored to balance false positives and false negatives: (1) setting an additional (albeit low) threshold on the prediction score; however, this could still make balancing false positives and false negatives difficult given the natural skew towards negative interactions in *Klebsiella* phage-host pairs; or (2) considering a flexible top-*k* ranking that would depend on the KL-type. For KL-types for which the model is very accurate, a smaller top-*k* can be considered (resulting in fewer false positives), while for KL-types that are more difficult to predict accurately, a larger top-*k* can be tested in the laboratory.

Importantly, we have not explicitly evaluated PhageHostLearn at the level of individual RBPs and their ability to bind to a specific K-antigen, as RBP-level validated interactions are unavailable. While some phages in the dataset consist of only a single RBP, and we would expect the model to learn these direct relationships between RBPs and their interacting K-antigen, most phages consist of two or more RBPs. For this reason, we do not know how accurate the model is in predicting KL-type specificity at the level of individual RBPs, nor can we assess how useful the model would be in assisting RBP engineering efforts to adjust the host range. These application settings can be further explored to broaden the model's usefulness.

More generally, PhageHostLearn represents a specific approach to predicting phage-host interactions at the level of the initial recognition of bacterial receptors by phage RBPs. Several other approaches exist for predictions at the species, genus, or higher levels, which are useful within different contexts[8,9]. For *Klebsiella* phage-host interactions, we argue that focusing on RBPs and their interacting K-antigens is appropriate and useful. However, phage-host pairs that do not interact with the K-antigen will currently be missed. Moreover, we have identified several phage defense systems in the clinical isolates, and the spot test results further suggest the importance of such defense systems. Due to the abundance of defense systems, we could not find clear correlations between defense systems and observed or absent phage-host interactions. Meanwhile, the presence of these systems underscores the relevance of also including such systems as information in machine learning models, for which we have proposed a multi-layer machine learning approach earlier[18]. PhageHostLearn is developed to serve as the first layer in that approach. Alternatively, high-capacity deep learning models might provide another way of modeling the infection process in its entirety, providing that sufficient amounts of data are available to train these complex models.

In summary, this work represents a first-of-its-kind approach that demonstrates the feasibility of predicting phage-host interactions at the strain level, given a comprehensive dataset of interacting phage-host pairs and their genomes. Moreover, the PhageHostLearn system is actionable and is evaluated in a practical setting. In that way, we believe PhageHostLearn meaningfully contributes to ongoing efforts in the machine-learning-guided development of phage therapeutics and diagnostics.

## Methods

### Sequence data collection and processing
Phage genome sequence data, bacterial genome sequence data and their in vitro verified interactions were collected from the Institute for Integrative Systems Biology (I²SysBio) in Valencia, Spain, as described by Beamud et al.[3]. This included collecting 138 clinical *Klebsiella* strains

from the Valencia region, spanning 59 different KL-types. These data were supplemented by an additional collection of phage-host interaction data for phages isolated on *Klebsiella* spp. reference strains that were sequenced with Illumina sequencing[26]. For both data sets, spot tests were carried out before in triplicate to verify *Klebsiella* phage-host interactions at a tenfold phage dilution, reflecting an initial interaction between phage RBPs and host receptors (but not necessarily a productive replication). In addition to the spot tests, Beamud et al.[3] further confirmed phage-host interactions with positive spot tests using a planktonic killing assay, measuring bacterial growth inhibition at $OD_{600nm}$ for at least 16 h.

Phage genome sequences were processed in three consecutive steps. In the first step, PHANOTATE was used to identify the genes in each phage genome[27]. Genes were identified without the use of tRNAscan-SE[40]. The second step involved translating the phage genes into proteins and detecting phage RBPs among them, for which we followed the method outlined in Boeckaerts et al.[28]. Briefly, this detection involves (1) computing HMM bit scores for each of the phage proteins against a manually curated set of RBP-related HMMs, (2) computing ProtBert-BFD embeddings for each of the proteins and (3) using both the bit scores and embeddings together as numerical representations in an XGBoost classifier that discriminates phage RBPs from other phage proteins. The code for this method was made publicly available. Finally, the third step in processing phage genomes involved discarding detected phage RBPs shorter than 200 amino acids and longer than 1500 amino acids, which is the range in length in which we expect RBPs, based on Latka et al. (2019)[29].

The bacterial genome sequences were processed with Kaptive[30,31]. More specifically, Kaptive was used to identify the capsule synthesis locus (K-locus) in each of the bacterial genomes using BLASTN against published K-locus reference sequences. The coding genes in each detected K-locus were translated into protein sequences and stored for further transformation into numerical features. When Kaptive detected missing genes, the corresponding reference gene of the best-matching KL-type was used for further processing. All the code for these analyses and processed data are made available through GitHub (https://github.com/dimiboeckaerts/PhageHostLearn) and Zenodo (code: https://doi.org/10.5281/zenodo.11074747; data: https://doi.org/10.5281/zenodo.11061100).

### Multi-instance feature representations
Phage RBPs and bacterial K-locus proteins were transformed into combined numerical vector representations (so-called features), representing both the phage and the bacterium together. We computed multi-instance representations using the pre-trained ESM-2 protein language model (t33_650M_UR50D configuration) that takes a single protein sequence as input and outputs a 1280-dimensional real vector that represents the protein[33]. Using ESM-2, each of the RBPs and K-locus proteins was transformed into a unique numerical vector. Next, the vectors of the RBPs corresponding to the same phage were averaged into a multi-instance representation for each phage. In the same way, the vectors of the K-locus proteins corresponding to the same bacterium were averaged into a multi-instance representation for that bacterium. Finally, these two multi-instance representations were concatenated into a combined 2560-dimensional vector representing a phage-host pair. These combined multi-instance feature representations then served as input for our machine learning model to learn interactions between known phage-host pairs and predict new interactions.

### A classification model that predicts interactions
The ESM-2-based feature representation was used as an input to train a binary XGBoost classifier. XGBoost is a nonlinear machine learning method that sequentially fits decision trees to improve the overall performance of the ensemble model. It is widely used for its broad

applicability and performance on unstructured data[24]. Three hyper-parameters were tuned using stratified five-fold cross-validation: the maximum depth of each tree (which influences the complexity of the model), the learning rate (which controls the optimization process), and the number of estimators (which refers to the number of boosting rounds that are done).

## In silico evaluation of the model

We have simulated the practical setting of finding candidate phages to test against a host of interest in silico by using our model to construct rankings of the predicted phages based on their prediction score. Then, we computed two metrics to evaluate our model performance in this setting: the mean hit ratio @ $k$ and the ROC AUC. The mean hit ratio @ $k$, defined as the probability of finding at least one matching phage in the top $k$, provides a meaningful visualization of the average probability of finding at least one hit in the top-$k$ candidates suggested by the model. More specifically, we have iteratively held out a single or group of bacterial genome(s) and its interactions at a time from the training set. After training, the model predicted the held-out interactions (consisting of different phages for one or more bacterial strains), and the prediction scores were used to construct rankings. We used these rankings to compute the hit ratio by comparing the top-$k$ ranked predictions to their ground truth label. This process was repeated for values of $k$ ranging from 1 to the total number of phages, resulting in a value for hit ratio for each of the values of $k$. Finally, this process was repeated for each of the groups of bacterial genomes in the dataset, iteratively taking out one group at a time, training the model on all the remaining data, and computing the hit ratio @ $k$ for the constructed rankings. All these values for hit ratio were then averaged across the number of bacterial genomes to produce a final mean hit ratio @ $k$, reflecting the average probabilities of finding hits in the top $k$ candidates suggested by the model. Practically, we accomplished this evaluation by implementing a leave-one-group-out cross-validation scheme (LOGOCV) in which each group represents one or more bacterial genomes and its associated interactions that were iteratively held out one by one for testing. Pairwise identity scores between K-locus nucleotide sequences were used to construct groups at set thresholds ranging from 75% to 100% (with identical sequences being grouped in either training set or test set at 100%). We have also simulated an 'informed microbiologist' approach, in which phages that are known to infect the same KL-type were prioritized to construct the ranking. If such phages were found for a given bacterium at hand, the order of suggested phages was further prioritized based on the number of other KL-types they additionally infect, *i.e.*, prioritizing narrow host range phages, as they typically exhibit a higher fitness[41]. Conversely, if no phages were found that infect the same KL-type, the broadest host range phages were prioritized.

In addition, we computed and visualized the ROC curve and computed its area-under-the-curve (AUC) in the same LOGOCV, without constructing a ranking. The ROC AUC can be interpreted as the probability that the model will score a randomly chosen interacting phage-host pair higher than a randomly chosen phage-host pair that does not interact.

## In vitro validation of the model with spot tests

A total of 28 currently circulating and carbapenem-resistant *Klebsiella pneumoniae* clinical isolates were collected and sequenced with Illumina in collaboration with the National Microbiology Center (CNM) in Madrid, Spain. Bacteria were isolated in several Spanish hospitals from samples of urine, blood, abscess, wound, ulcer, and rectal exudates. The bacterial genomes were sequenced with Illumina sequencing. Afterwards, the bacterial genome sequences were processed with Kaptive as before. Each of these genomes was used as an input into PhageHostLearn to predict interactions and construct a ranking for the collection (I²SysBio) of 59 phages isolated on *Klebsiella* spp. reference strains and for which the full genome was available. The top-five ranked phages for each *K. pneumoniae* clinical isolate were validated in the laboratory using spot tests in semi-solidified media at 1:10 and 1:10³ phage dilutions (in liquid broth) in duplicate or triplicate (for discrepant results). First, bacterial cultures were inoculated from glycerol stocks and grown overnight at 37 °C in liquid broth. Phage stocks were aliquots of an amplification in liquid broth and stored at −80°. Spots were done by adding drops of 1 µl at 1:10 or 1:10³ phage dilution to bacterial lawns of 200 µL of each of the 28 *K. pneumoniae* isolates and 3.5 mL of 0.3% LB top agar in Petri plates. To assure the quality of the phage stocks, each phage was also tested on its isolation strain at 1:10 and 1:10³ phage dilutions as positive controls. Plates were incubated for 24 h at 37 °C.

A spot was considered positive at a certain dilution if either clear plaques were observed or cases in which it was not possible to distinguish clear plaques were observed (potentially indicating lysis from without but a positive RBP-receptor interaction), both for phages tested against the clinical isolates and the positive controls. The absence of spots was considered a negative result. At least two replicates of the experiment were performed, and a third replicate was performed if discrepancies were observed. In those cases, the final result was negative if spots for at least two replicates were absent and positively scored if at least spots for two replicates were confirmed at a certain dilution. Laboratory confirmations were used to visualize both the hit ratio @ $k$ and ROC AUC in the same way as before. Finally, DefenseFinder was used to find phage defense systems in the isolates' genomes[34].

The biological materials used in the in vitro validation of this study are available from the Institute for Integrative Systems Biology (I2SysBio, contact: pilar.domingo@uv.es) and the Spanish Microbiology Center (CNM, contact: jesus.oteo@isciii.es) under a data use agreement upon request.

## Inclusion and ethics statement

This research is a collaboration between Belgian and Spanish researchers related to five different research groups. The research is especially relevant in both countries, focusing on *Klebsiella pneumoniae*, a highly relevant pathogen in clinical settings. *Klebsiella* is by extension also globally relevant including in low- and middle-income countries. The research made use of currently circulating and carbapenem-resistant *Klebsiella pneumoniae* clinical isolates, working together with the National Microbiology Center (CNM) in Madrid, Spain. Responsibilities for this research were agreed amongst collaborators ahead of the research. Where possible, local relevant research was taken into account in citations.

## Reporting summary

Further information on research design is available in the Nature Portfolio Reporting Summary linked to this article.

# Data availability

All data used and generated in this study have been deposited in Zenodo under accession code 11061100. These data include (1) the raw sequence data collected in FASTA format from Beamud et al.[3]. and from Ferriol-González et al.[26]; (2) the processed data that were used in the analyses and to train and evaluate the machine learning model and (3) the phage-host interaction data in a.csv format.

# Code availability

We provide full availability of our code through GitHub (https://github.com/dimiboeckaerts/PhageHostLearn) and Zenodo (https://doi.org/10.5281/zenodo.11074747). Sequence data were processed using PHANOTATE v1.5.0 (https://github.com/deprekate/PHANOTATE), PhageRBPdetection v2.1.3 (https://github.com/dimiboeckaerts/PhageRBPdetection) and Kaptive v2.0.0 (https://github.com/

klebgenomics/Kaptive). Feature representations of processed sequences were computed using ESM-2 v1.0.3 (https://github.com/facebookresearch/esm). The machine learning model used XGBoost v1.5.0 (https://github.com/dmlc/xgboost) and we evaluated the model using cross-validation and metrics implemented in Scikit-learn v0.24.2 (https://scikit-learn.org/stable/). Furthermore, our code pipeline uses python v3.9.7, biopython v1.79, joblib v1.1.0, json v4.2.1, matplotlib v3.4.3, numpy v1.20.3, pandas v1.3.4, pickle 0.7.5 and seaborn v0.11.2.

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

## Acknowledgements

D.B. is supported by the Research Foundation – Flanders (FWO), grant number 1S69520N. M.S. and B.D.B. received funding from the Flemish Government under the "Onderzoeksprogramma Artificiële Intelligentie (AI) Vlaanderen" program. Project PID2020-112835RA-I00 funded by MCIN/AEI /10.13039/501100011033, and project SEJIGENT/2021/014 funded by Conselleria d'Innovació, Universitats, Ciència i Societat Digital (Generalitat Valenciana) to P.D-C. P.D-C. was financially supported by a Ramón y Cajal contract RYC2019-028015-I funded by MCIN/AEI/ 10.13039/501100011033, ESF Invest in your future.

## Author contributions

Conceptualization & Methodology: D.B., M.S., B.D.B., R.S., P.D.-C., and Y.B.; Data Curation, Software, Formal Analysis, Validation & Original Draft preparation: D.B.; Experimental design: D.B., M.S., R.S., P.D.-C., B.D.B., and Y.B. Experimental validation: C.F.-G.; Molecular characterization of *K. pneumoniae* high-risk clones: J.O.-I.; Manuscript Review & Editing: D.B., M.S., B.D.B., R.S., P.D.-C., and Y.B.; Supervision: M.S., B.D.B., R.S., P.D.-C., and Y.B.

## Competing interests

The authors declare no competing interests.
