## [Peer Review File · Nature Communications]

Prediction of Klebsiella phage-host specificity at the strain levelEditorial Note: This manuscript has been previously reviewed at another journal that is not operating a transparent peer review scheme. This document only contains reviewer comments and rebuttal letters for versions considered at *Nature Communications*.

Reviewer #1 (Remarks to the Author):

This manuscript is a revised version of work by Boeckaerts et al. from Yves Briers' group + collaborators and describes PhageHostLearn, a new tool that predicts phage host recognition for Klebsiella.

I remain very positive about the quality of the work, its depth, and its presentation. Furthermore, I thank the authors for their thoughtful and interesting response to my comments about the earlier version of this manuscript. My comments have been implemented via textual changes, but the authors have not widened the limited scope and breadth of their work. It therefore essentially remains limited to predicting which Klebsiella phage can target which capsule type of Klebsiella, and it does so quite well with interesting results. However, as outlined by the authors themselves in their response, this work can not be applied to other bacteria or for a true prediction of Klebsiella phage host range beyond initial capsule recognition. This makes me doubtful about the broad relevance of this work and whether the predictions of PhageHostLearn are as "actionable" as indicated by the manuscript title. I have no further comments or concerns to add to those about the earlier version of this manuscript.

Reviewer #4 (Remarks to the Author):

Thank you for taking the time to make the adjustments to the manuscript. All of the queries that I had in the previous review have been answered, and for the others reviewers too.

I maintain that this is a very useful tool and has potential for many projects, which has clearly been demonstrated in the provided comments.

Reviewer #6 (Remarks to the Author):

The manuscript titled "Actionable prediction of Klebsiella phage-host specificity at the strain level" by Boeckaerts et al. introduces a novel method for predicting host-phage relationships using genomic sequences from Klebsiella and phages. The authors present innovative concepts and address an important problem centered on the prediction of complex host-phage interactions. The manuscript is well-written and demonstrates clear relevance to phage therapy research. However, I am concerned that the current benchmarks may not be adequate to confirm the model's ability to generalize to new strains, or to demonstrate that PhageHostLearn can accurately predict strain-specific phages as opposed to phages with a broad host range.

Major comments:

- My main concern is that the authors do not properly show whether independence between the train and test set was ensured. If very similar host-phage pairs are found in both the train and the test sets, such data leakage would inflate the model's performance and wouldn't allow confirmation that the model generalizes well for pairs that are not very similar to anything in the train data. The authors do perform a leave-one-group-out cross-validation scheme (LOGOCV), but this approach is not sufficient if the dataset contains groups of very closely related strains, as even when a genome is excluded from the train data, other genomes that are very similar to it might be included in it. Therefore, I suggest that the authors (1) measure pairwise similarities of genomes in the data; and (2) perform model evaluation in such a way that groups of very similar genomes are excluded from the dataset together (that is, groups of genomes would be treated as a unit during LOGOCV, rather than a single genome). Although it is impossible to guarantee complete independence of train and test sets in biological data, given that all genomes are related, the

authors could use the distance of the K-locus proteins embeddings as proxies to genomic distances and cluster together genomes that have similar embeddings.

- Lines 246-259: This is an interesting result that raises the question: how does PhageHostLearn improves over just picking broad host range phages? To answer this the authors could evaluate (1) how frequently PhageHostLearn correctly ranks a KL-type specific phage over a broad host range phage, and (2) how much variance do you get for the predictions of bacterial genomes of different KL-type (if PhageHostLearn mostly finds broad host range, we would expect little variance).

- Lines 233-235: The authors mention that defense systems could be the underlying reason for the numbers they present, but this is not further explained. Why would defense systems explain those results? In the discussion, the authors say that they couldn't find a correlation between defense systems and false positive predictions, but this is not presented with data. Even though incorporating defense system information is out of the current scope of PhageHostLearn, the authors should present the data that suggest that the presence of defense systems explain (or do not explain) deviations from PhageHostLearn predictions.

Minor comments:

- In line 175, it's not clear that the hit ratio corresponds to the chance of finding at least one matching phage. My first impression is that it represented the proportion of matching phages among the top-k predictions, which would result in a descending curve in panel 2A, not ascending. Please describe the hit ratio metric clearly once it is first introduced.

- Lines 231 and 232: "from a machine learning perspective" What does this mean?

- Could the authors provide the area under the precision-recall curve in addition to the ROC AUC:

Code and reproducibility:

- The notebook in the repository are very detailed and well written. However, if a researcher wants to run PhageHostLearn on their data, it's not entirely clear what they should do, as files are spread across directories and no step-by-step guide is available. I suggest that the authors provide a short tutorial on how to execute the code in the README, otherwise the value to the community is limited.

- Have the authors checked if the notebooks can be executed and reproduced? I noticed, for instance, that the "compute_protein_embeddings" function is commented out, which precludes the inference of the code on a new dataset.

Reviewer #6 (Remarks on code availability):

As mentioned in my review. The code is high-quality, but the authors should improve the documentation to allow users to run inference on their own data and also check if the notebooks can be executed.

Response to reviewers

Prediction of *Klebsiella* phage-host specificity at the strain level

We sincerely thank the reviewers for providing us a second round with valuable feedback. Their comments and our point-by-point responses to address their comments are elaborated below.

Reviewer #1

This manuscript is a revised version of work by Boeckaerts et al. from Yves Briers' group + collaborators and describes PhageHostLearn, a new tool that predicts phage host recognition for *Klebsiella*.

I remain very positive about the quality of the work, its depth, and its presentation. Furthermore, I thank the authors for their thoughtful and interesting response to my comments about the earlier version of this manuscript. My comments have been implemented via textual changes, but the authors have not widened the limited scope and breadth of their work. It therefore essentially remains limited to predicting which *Klebsiella* phage can target which capsule type of *Klebsiella*, and it does so quite well with interesting results. However, as outlined by the authors themselves in their response, this work cannot be applied to other bacteria or for a true prediction of *Klebsiella* phage host range beyond initial capsule recognition. This makes me doubtful about the broad relevance of this work and whether the predictions of PhageHostLearn are as "actionable" as indicated by the manuscript title. I have no further comments or concerns to add to those about the earlier version of this manuscript.

We thank the reviewer for their continued enthusiasm and compliments on the quality of our work. We also understand the remaining concern of the reviewer about our particular focus on *Klebsiella*, resulting in a narrow scope of the work, which is primarily due to a current lack of available data for other important bacterial species. We have clearly mentioned this in our manuscript (*Lines 82–85*) and have also additionally clarified this in the first round of revision (*Lines 290–298*). We agree with the reviewer's doubts about the actionability of our predictions, which we initially proposed in order to reflect our actionable output format compared to other published work. We have removed the term 'actionable' from the title to avoid ambiguity in interpretation. We also adjusted our phrasing in the discussion accordingly:

Lines 289–290:

*"We specifically trained and evaluated our system to make predictions for *Klebsiella* phage-host pairs in an actionable format".*

We would like to emphasize that moving host prediction to the strain level substantially shifts the current state-of-the-art. The approach we pursued is inherently generic, allowing for its application across various species. As more phage-host interaction data become widely available, our methods can be readily extended to encompass additional species. By doing so, we anticipate further improvements in accuracy and applicability in the future.

Reviewer #4

Thank you for taking the time to make the adjustments to the manuscript. All of the queries that I had in the previous review have been answered, and for the other reviewers too. I maintain that this is a very useful tool and has potential for many projects, which has clearly been demonstrated in the provided comments.

We appreciate the reviewer's kind words on the importance of our work.

Reviewer #6

The manuscript titled "Actionable prediction of Klebsiella phage-host specificity at the strain level" by Boeckaerts et al. introduces a novel method for predicting host-phage relationships using genomic sequences from Klebsiella and phages. The authors present innovative concepts and address an important problem centered on the prediction of complex host-phage interactions. The manuscript is well-written and demonstrates clear relevance to phage therapy research. However, I am concerned that the current benchmarks may not be adequate to confirm the model's ability to generalize to new strains, or to demonstrate that PhageHostLearn can accurately predict strain-specific phages as opposed to phages with a broad host range.

We thank the reviewer for the positive words on our work and have tried to accommodate to every comment below.

Major comments:

- My main concern is that the authors do not properly show whether independence between the train and test set was ensured. If very similar host-phage pairs are found in both the train and the test sets, such data leakage would inflate the model's performance and wouldn't allow confirmation that the model generalizes well for pairs that are not very similar to anything in the train data. The authors do perform a leave-one-group-out cross-validation scheme (LOGOCV), but this approach is not sufficient if the dataset contains groups of very closely related strains, as even when a genome is excluded from the train data, other genomes that are very similar to it might be included in it. Therefore, I suggest that the authors (1) measure pairwise similarities of genomes in the data; and (2) perform model evaluation in such a way that groups of very similar genomes are excluded from the dataset together (that is, groups of genomes would be treated as a unit during LOGOCV, rather than a single genome). Although it is impossible to guarantee complete independence of train and test sets in biological data, given that all genomes are related, the authors could use the distance of the K-locus proteins embeddings as proxies to genomic distances and cluster together genomes that have similar embeddings.

This is a great remark by the reviewer. We initially decided not to pay specific attention to how far the model can generalize. Instead, we focused on generating and assembling a diverse training dataset that covers the majority of *Klebsiella* KL types and subsequently testing our model on novel clinical isolates. However, we realize that this is nonetheless a valuable evaluation to include in our results. For this reason, we computed pairwise alignment scores between the loci nucleotide sequences and reran our leave-one-group-out cross-validation, considering the alignment scores to guide grouping and exclude similar loci from the same validation split. Indeed, this makes it more challenging for the model to make predictions, which is reflected in lower performances (ROC AUC scores from 65.7% to 81.8%), although the decrease in performance is not always consistent across decreasing thresholds for alignment scores. We adjusted the results in Figure 2 accordingly. We continue to be confident about our *in vitro* evaluation as a fair evaluation of how well the model works in practice.

Corresponding to the adjusted results, we have made adjustments in the Results and Methods sections of the manuscript:

Lines 169–173:

"We have simulated this representative setting in silico by iteratively holding out a single or group of bacterial genome(s) with its phage interactions at a time from the training set. Pairwise identity scores

between K loci nucleotide sequences were used to construct groups at set thresholds ranging from 75% to 100%.”

Lines 181–184:

“For example, with our model we expect to find at least one hit in the top-10 in around 65% to 84% of the cases on average. Its performance decreases as the model makes predictions for bacterial strains that are increasingly dissimilar to those seen in training (blue-green curves).”

Line 187:

“Our model achieves a comparable performance compared to that of an informed microbiologist.”

Line 192:

“Our model reaches ROC AUC values between 65.7% and 81.8%.”

Lines 201–203:

“Figure 2: a. Mean hit ratio @ k of the trained XGBoost model in a LOGOCV at decreasing thresholds for K loci identity (blue-green curves) and of an informed microbiologist approach (red). b. ROC curve with AUC of the trained XGBoost model in a LOGOCV at decreasing thresholds for K loci identity.”

Lines 448–451:

“Finally, this process was repeated for each of the groups of bacterial genomes in the dataset, iteratively taking out one group at a time, training the model on all the remaining data and computing the hit ratio @ k for the constructed rankings.”

Lines 453–458:

“Practically, we accomplished this evaluation by implementing a leave-one-group-out cross-validation scheme (LOGOCV) in which each group represents one or more bacterial genomes and its associated interactions that were iteratively held out one by one for testing. Pairwise identity scores between K loci nucleotide sequences were used to construct groups at set thresholds ranging from 75% to 100%.”

- Lines 246-259: This is an interesting result that raises the question: how does PhageHostLearn improves over just picking broad host range phages? To answer this the authors could evaluate (1) how frequently PhageHostLearn correctly ranks a KL-type specific phage over a broad host range phage, and (2) how much variance do you get for the predictions of bacterial genomes of different KL-type (if PhageHostLearn mostly finds broad host range, we would expect little variance).

We don't explicitly claim that PhageHostLearn will predict narrow host range phages better than broad host range phages, although one could definitely presume this based on our focus on phage RBPs and K-loci proteins. The broader host range phages indeed occur across the true positives, false positives and false negatives (*Lines 249–252*). We hypothesize that there can be some signal in the representations of the phage RBPs that is learned by the model to predict broad host range phages, even though the signal from the concatenated K-loci proteins representation might be absent or even conflicting, resulting in both good and bad predictions for these broader host range phages. We kindly disagree with the reviewer that it would be interesting to validate whether our model ranks narrow host range phages higher than broader host range phages because we did not formulate our model in any way to explicitly learn such differences. However, we believe that an interesting, related metric to look at is how often a narrow host range phage was missed while a broad host range phage was found (given that at least one narrow host range phage was experimentally confirmed). It would indeed be desirable for the model to find both the narrow host range phages and the broader host range phages. Out of the 16 clinical strains for which we confirmed one or more interactions in the lab, we were able to confirm an interaction with a narrow host range phage for six of these strains. In five of these cases,

our model also correctly ranked those narrow host range phages in the top 5 that were evaluated in the lab. We have added this result to our manuscript:

Lines 255–257:

“The model also correctly suggests KL-type specific phages (e.g., K54λ1.1.1 and K17α62) for five out of the six clinical strains for which KL-type specific phages could be confirmed experimentally (Supplementary material S3).”

- Lines 233-235: The authors mention that defense systems could be the underlying reason for the numbers they present, but this is not further explained. Why would defense systems explain those results? In the discussion, the authors say that they couldn't find a correlation between defense systems and false positive predictions, but this is not presented with data. Even though incorporating defense system information is out of the current scope of PhageHostLearn, the authors should present the data that suggest that the presence of defense systems explain (or do not explain) deviations from PhageHostLearn predictions.

In our manuscript at *Lines 231-233*, we are specifically referring to the observed differences in confirmed interactions between phages and hosts at a 1:10 dilution and a 1:10³ dilution. If we assume that lysis from without can occur at a 1:10 dilution but not at a 1:10³ dilution, then a potential explanation for the differences in confirmed interactions at these two dilutions is that phages can attach to the host surface but not successfully introduce and activate their genomes into their host because of phage (secondary) defense systems. We clarified this in the text:

Lines 233–235:

“These differences between the two dilutions may indicate the presence of one or more active phage defense systems (due to the occurrence of lysis without successful replication at a 1:10 phage dilution).”

In our discussion, we mentioned that we did not find clear correlations between infection patterns and defense systems (*Lines 364–366*), which is not the same as what the reviewer mentions (false positives). What we mean to say is that we observe an abundance of detected defense systems, impeding us from manually identifying clear patterns in how such defense systems would be linked to an experimental confirmation or absence of the observed phage-host interactions. We have rephrased our text accordingly:

Lines 364–366:

“Due to the abundance of defense systems, we could not find clear correlations between defense systems and observed or absent phage-host interactions.”

Minor comments

- In line 175, it's not clear that the hit ratio corresponds to the chance of finding at least one matching phage. My first impression is that it represented the proportion of matching phages among the top-k predictions, which would result in a descending curve in panel 2A, not ascending. Please describe the hit ratio metric clearly once it is first introduced.

We extended the following sentence to clarify our definition of the hit ratio:

Lines 174–177:

“The hit ratio, defined as the probability of finding at least one matching phage, was computed across the top-k ranked phages by comparing the ranked predictions to the ground truth labels to quantify how well our model finds matching phages.”

- Lines 231 and 232: “from a machine learning perspective” What does this mean?

We rephrased the following sentences to clarify our explanation:

Lines 229–231:

“One or more interactions were confirmed with spot tests for 16 out of the 28 bacterial isolates. Out of these 16 isolates, one or more interactions could be confirmed at both dilutions for 12 of the isolates (Supplementary material S3).”

- Could the authors provide the area under the precision-recall curve in addition to the ROC AUC:

This is a great remark by the reviewer. We have spent a significant amount of time discussing and thinking about the appropriate evaluation metrics for our machine learning model. Although we agree that, most often, a PR curve would be a great addition to the evaluation, we believe that in this particular setting, the PR curve can be misleading. The difference between the ROC and PR curves essentially boils down to the difference between the Precision (TP/TP+FP) and the False Positive Rate (FP/FP+FN). In our manuscript, we argue that we can tolerate False Positives to some extent and find it more important to avoid False Negatives (*Lines 329–334*). Especially given the very low number of True Positives overall (2.74%), we believe that Precision is not a good metric, as it focuses on the True Positives in relation to the False Positives. Consider the hypothetical case in which ten interactions out of fifty are predicted as positive, among which one true positive. While the Precision would only be 10% (1/10) here, we would still consider this a successful result of our model, as we correctly find the one TP and reduce laboratory validations from 50 to 10.

For these reasons, we have chosen to present the ROC curve and its AUC as an additional evaluation metric, but we also want to emphasize that the most important evaluation metric in our opinion is the hit ratio @ *k* curve.

Code and reproducibility:

- **The notebooks in the repository are very detailed and well written. However, if a researcher wants to run PhageHostLearn on their data, it’s not entirely clear what they should do, as files are spread across directories and no step-by-step guide is available. I suggest that the authors provide a short tutorial on how to execute the code in the README, otherwise the value to the community is limited.**
- **Have the authors checked if the notebooks can be executed and reproduced? I noticed, for instance, that the “compute_protein_embeddings” function is commented out, which precludes the inference of the code on a new dataset.**

We thank the reviewer for these final comments, as we are heavily committed to making our tool available and easy to use for researchers. First, we have improved and simplified the *phagehostlearn_inference.ipynb* notebook which researchers can use to make predictions for their own data. Second, we have expanded the README file in the repository with two sets of clear steps to follow for researchers that either (1) want to make predictions for their own data or (2) reproduce our analyses. Before, we provided only one combined set of steps, which was less clear. Third, we double-checked all the code in the notebooks and utility scripts and made corrections where needed. Specifically, we had indeed commented out the ‘*compute_protein_embeddings*’ function due to a technical defect on a local computer, but this is indeed a function that is required to complete the predictions (researchers can either run this locally or in the cloud with the provided *PTBembeddings_cloud.ipynb* notebook). All the updated code and information can be found in our GitHub repository: <https://github.com/dimiboekaerts/PhageHostLearn>.

Reviewer #6 (Remarks on code availability):

As mentioned in my review. The code is high-quality, but the authors should improve the documentation to allow users to run inference on their own data and also check if the notebooks can be executed.

We provided the answer to this comment in the comment above.

Reviewer #6 (Remarks to the Author):

I appreciate that the authors addressed my main comments and provided satisfactory responses. However, I remain concerned about the following statement: "Our model reaches a comparable performance to that of an informed microbiologist."

According to panel 2A, this claim appears to hold true only when there is data leakage in the training process, meaning the same K-locus sequences are present in both the training and test sets. Once the authors ensure that sequences in the training and test sets are at most 99.5% identical (i.e., have at least one difference every 200 nucleotides), the naive approach of "selecting from a subset of phages that are known to infect the same KL-type as the bacterial strain at hand" outperforms their model. This suggests that the model's superior performance compared to the naive classification was likely due to redundancy between the training and test sets, and minimal differences were sufficient to significantly impact the model's performance.

If my interpretation is correct, the authors should be more cautious with their wording and tone down the model's performance claims throughout the paper, taking into account that the model doesn't seem to generalize well.

Could the authors clarify what is their interpretation of the fact that the naive classifier outperformed their model when data leakage was controlled for? While panel 2A seems to indicate that some performance claims are misleading, this crucial result is not adequately discussed in the manuscript. A thorough discussion of this result would ensure that the model's capabilities and limitations are accurately represented.

Response to reviewers

Prediction of *Klebsiella* phage-host specificity at the strain level

We sincerely thank the reviewer for providing us valuable final comments. We have addressed the feedback below.

Reviewer #6

I appreciate that the authors addressed my main comments and provided satisfactory responses. However, I remain concerned about the following statement: "Our model reaches a comparable performance to that of an informed microbiologist."

According to panel 2A, this claim appears to hold true only when there is data leakage in the training process, meaning the same K-locus sequences are present in both the training and test sets. Once the authors ensure that sequences in the training and test sets are at most 99.5% identical (i.e., have at least one difference every 200 nucleotides), the naive approach of "selecting from a subset of phages that are known to infect the same KL-type as the bacterial strain at hand" outperforms their model. This suggests that the model's superior performance compared to the naive classification was likely due to redundancy between the training and test sets, and minimal differences were sufficient to significantly impact the model's performance.

If my interpretation is correct, the authors should be more cautious with their wording and tone down the model's performance claims throughout the paper, taking into account that the model doesn't seem to generalize well.

Could the authors clarify what is their interpretation of the fact that the naive classifier outperformed their model when data leakage was controlled for? While panel 2A seems to indicate that some performance claims are misleading, this crucial result is not adequately discussed in the manuscript. A thorough discussion of this result would ensure that the model's capabilities and limitations are accurately represented.

We agree with the reviewer that this is an important result to discuss. In Figure 2a, the "LOGOCV @ 100 id%" means that we have discarded 100% identical sequences from the training and test sets, meaning that no identicals occur across the two sets. This might not have been clear from our explanation in the text. In that regard, we still feel it is fair to say that the model achieves a comparable performance as an informed microbiologist at that level of clustering, when both the machine learning model and the microbiologist approach receive exactly the same information in the comparison. For lower thresholds of clustering, the microbiologist can outperform the machine learning model, given the fact that the model indeed seems to be sensitive to small changes in similarity of the bacterial host. However, even a tool that does not reach human-level performance can still be very helpful to decrease labor intensive laboratory work. We have rewritten the text in a more considerate way:

Lines 22 – 23:

"PhageHostLearn reaches a cross-validated ROC AUC of up to 81.8% in silico and maintains this performance in laboratory validation."

Lines 188 – 191:

"PhageHostLearn can reach a performance comparable to that of an informed microbiologist at the highest threshold for grouping, however, given the decrease in performance for only slightly lower thresholds, seems to rely on very similar K-locus sequences to reach that level of performance."

Lines 206 – 208:

“At the 100% threshold for grouping, identical K-locus sequences are grouped together, either in the training set or test set.”

Lines 449 – 451:

“Pairwise identity scores between K-locus nucleotide sequences were used to construct groups at set thresholds ranging from 75% to 100% (with identical sequences being grouped in either training set or test set at 100%).”

Furthermore, to the reviewer’s point of generalizability, we indeed observe that the model performance first drops at higher clustering thresholds, and then stabilizes at lower clustering thresholds around 90-95 id % (reaching a hit ratio @ 10 of around 68-70% and a ROC AUC of around 68%). Our explanation of these results is the following. In our dataset, the majority of *Klebsiella* phage-host pairs are KL-type specific interactions. Identical KL-types are typically defined as K locus sequences that are 90% or more identical. In that regard, the model does seem to learn some robust patterns that are KL-type independent, but currently overfits on KL-type specific patterns (which can be observed by the decrease in performance from 100-90 id %). We have clarified this in the Results section of our manuscript:

Lines 180 – 186:

“The model’s performance decreases as it makes predictions for bacterial strains that are increasingly dissimilar from those seen in training (blue-green curves) and stabilizes at thresholds around 90% identity, a typical threshold for defining unique KL-types [30]. Given a majority of KL-type specific phage-host interactions in our dataset, we presume that PhageHostLearn learns some robust patterns that are KL-type independent, but currently overfits on KL-type specific patterns (which can be observed by the decrease in performance from 100 to 90 % identity).”